



# Litter vs. Lens: Evaluating LAI from Litter Traps and Hemispherical Photos Across View Zenith Angles and Leaf Fall Phases

Simon Lotz[1], Teja Kattenborn[1], Julian Frey[1,2], Salim Soltani[1], Anna Göritz[1], Tom Jakszat[1], and Negin Katal[1]

[1]Sensor-based Geoinformatics, University of Freiburg, Germany
[2]Forest Growth and Dendroecology, University of Freiburg, Germany

**Correspondence:** Simon Lotz (simon.lotz@posteo.de)

**Abstract.** Leaf area index (LAI) is a key parameter for modeling ecosystem productivity, climate interactions, and hydrological processes, as well as monitoring vegetation health. While satellite-based estimates provide insights into large-scale vegetation dynamics, ground-based methods, including digital hemispherical photography (DHP), are essential to generate and validate such products and offer a practical alternative for fine-scale assessments. However, it remains unclear if the DHP method enables to robustly track temporal LAI dynamics. Here, we evaluate DHP-derived LAI time series with litter trap (LT)-derived LAI in a temperate deciduous broad-leaved forest. First, by comparing DHP-derived LAI estimates with LT-derived LAI across varying view zenith angles ranging from 10° to 90°, we investigate how well both methods align. Using 15 sample locations, we found the highest average correlation across all locations of DHP- and LT-derived LAI ($R^2$=0.88) at a view zenith angle of 20°, indicating that litter traps represent a relatively narrow spatial footprint. Uncertainties for individual litter traps attributed to varying site conditions, such as tree stem density or canopy coverage. To overcome these uncertainties, we applied a site specific calibration using the litter traps and a generalized linear mixed model, which significantly increased correlation ($R^2$=0.97).

This study highlights the potential of DHP for tracking spatio-temporal LAI dynamics in decideous forests. Moreover, we demonstrate that integrating DHP and LT data, alongside a mixed-effects model, can enhance the site specific accuracy and applicability of LAI assessments.

## 1 Introduction

Vegetation canopies regulate energy, water, and carbon fluxes at the biosphere-atmosphere interface, fundamentally influencing climate dynamics and agroecosystem functioning (Bonan, 2008). As leaves constitute the active surfaces for these mass and energy exchanges, the leaf area index (LAI) emerges as a fundamental biophysical parameter - quantifying canopy structural complexity through its definition as half of the total intercepting area per unit ground surface area ($m^2/m^2$) (Chen and Black, 1992). LAI plays a critical role in ecosystem productivity models as well as global models of climate and hydrology influencing carbon, water and energy fluxes (Myneni et al., 2002). LAI is also estimated on large spatial and temporal scales from satellites to track plant phenology (Verger et al., 2016) and monitor vegetation health and productivity (Das et al., 2024). Various





methods and tools have been designed to estimate LAI, which is typically measured using either direct or indirect techniques (Bréda, 2003; Fang et al., 2019). Direct measurement of LAI remains methodologically challenging, restricting its precise

quantification at local scales and complicating the validation and calibration of satellite-based LAI retrievals across broader geographic regions. Direct methods involve destructive sampling, requiring the collection and measurement of all leaf material within a defined ground area (Baret et al., 2010). While this approach provides highly accurate LAI values, it is labor-intensive, costly, and impractical for long-term monitoring or conservation-sensitive environments. Consequently, indirect methods are more commonly employed, as they allow for repeated, cost-effective, and non-invasive LAI estimation. One indirect method

is the litter trap (LT) method, which involves deploying containers of known ground area to collect fallen leaves over time (Černỳ et al., 2019). The ratio of the total leaf area accumulated within the trap to its ground area provides an estimate of LAI. Although LTs are considered one of the most accurate methods for retrieving LAI they are less effective for multitemporal analysis. This is because each time step requires emptying the traps and measuring the leaf surfaces, a process that is highly labor-intensive. Moreover, they cannot be used to retrieve the LAI of evergreen plants (Jonckheere et al., 2004). Another

limitation is that the spatial representativeness of LTs often remains unknown and may vary with tree height, LT size and wind conditions (Schaefer et al., 2015). An alternative non-destructive LAI retrieval is given by indirect optical measurement techniques because of their non-destructive, time-effective and flexible application. An example are photosynthetically active radiation (PAR)-based LAI estimations, which rely on measuring the attenuation of PAR (400–700 nm) above and below the canopy. The relative difference in PAR transmission is used to estimate LAI via the Beer-Lambert Law, which relates light

attenuation to leaf area (Saitoh et al., 2012). However, PAR-based methods assume that plant canopies are a homogenous medium. In complex canopies, the accuracy of PAR-based methods is constrained due to light scattering, foliage clumping, and saturation at high LAI values (Yao et al., 2016). Accordingly PAR-based methods were found to be inadequate for forests due to the complexity of canopy structure, including branches and stems, leaf clumping and leaf angle variation (Bréda, 2003; Chen and Black, 1992; Geng et al., 2021).

Another indirect method for estimating LAI is using digital hemispherical photography (DHP), which captures wide-angle images of the canopy from beneath using a fisheye lens (Chianucci and Cutini, 2013). Such images can be taken at various sample points within the study area and provide detailed information on canopy gaps, leaf density, and stem distribution. Additionally, DHP helps assessing canopy complexity, including clumping effects and the presence of stems. The DHP methods enable segmentation of the image into zenith rings and azimuth segments, allowing for detailed assessment of gap fraction and

LAI at various zentih angles. This approach provides insights into canopy architecture and light transmission while accounting for leaf clumping and the mean leaf inclination angle, enhancing the accuracy of LAI estimation (Chianucci and Cutini, 2012). To correct for non-random spatial distribution of foliage within plant canopies clumping indices are applied, correcting for the underestimation of true LAI in traditional estimation methods (Chen and Black, 1992). Early DHP methods underestimated LAI by up to 44–70% compared to LT references, primarily due to unaccounted canopy clumping effects (Jonckheere et al.,

2004; Zhang et al., 2005). The introduction of modern clumping index corrections, such as the LXG method by Chianucci et al. (2019) and CLX approach by Leblanc et al. (2005), has significantly improved accuracy, reducing errors to <15-21% in mixed forests (Chianucci, 2020). However, the accuracy of DHP methods was mostly evaluated with temporal snapshots and





not in temporally consistent setting. Hence, the robustness and consistency of DHP estimates for tracking LAI through time remains mostly unknown. Moreover, recent studies highlight persistent challenges in DHP-based LAI estimation, particularly
regarding view zenith angle (VZA) selection and effects of stem density and canopy structure (Lee et al., 2023). Lee et al. (2023) demonstrated that larger zenith angles (>60°) tend to overestimate gap fraction due to increased contributions from clumping effects and woody material, while narrower angles (<30°) may underestimate LAI by missing critical canopy details.

Here, we evaluate DHP-based LAI estimates with LAI derived from litter traps for decideous broadleaved trees in a temperate forest using a multitemporal setting (8 repeated measurements in 4 months). Thereby, we evaluate the LAI derived from the
DHP method using different VZA with LT-derived LAI. This evaluation not only helps to identify the most suitable VZA configuration for accurate DHP-based LAI estimation, but also provides insight into the spatial footprint and representativeness of litter traps in structurally heterogeneous canopies. Moreover, we show that systematic site-specific biases, e.g. due to the local canopy structure or woody components, can be overcome by a site-specific calibration of DHP-estimates with LT-derived LAI, ultimately improving the robustness of DHP-based LAI assessments.

## 2 Methodology

### 2.1 Study site and data acquisition

The data was collected on the site of ECOSENSE project, a multi-scale research initiative focused on quantifying and modeling spatiotemporal dynamics of ecosystem processes through sensor networks. The site is located in a mixed forest system dominated by beech with patches of conifer trees near Ettenheim, Germany (Werner et al., 2024). The site is located at 48.2679°N,
7.8783°E with elevation ranging from around 450 to 520 m a.s.l (Werner et al., 2024). The basal area of the forest site is 30.9 m$^2$/ha. The diameter at breast heights vary with a mean = 23 cm, median = 16 cm, standard deviation 15 cm and maximum of 76 cm.

The sample points for DHP and LT measurements were setup in a regular grid of 50 meters distance. Since litter traps are restricted to deciduous stands, sample points containing coniferous trees were excluded, resulting in 15 sample points (Figure
1b).The average canopy coverage of the sample points amounts to 97.6% and varies between 93.6% to 98.8% and average canopy heights of the sample points vary between 22 to 27 meters (Figure A3) . Multi-temporal LAI estimates with LTs and DHP were acquired before the start to the end of the leaf fall between September 20th to December 16th 2024 (Figure 2).

### 2.2 LAI Derived from litter traps

At each sample point, a LT (40x60x30 cm) was installed to systematically collect leaf litter (Figure 1d). Each LT was equipped
with a drainage fleece to prevent leaf loss.The trap height of 30 cm was chosen to prevent the escape of leaves due to wind, ensuring accurate collection within the box. We used a surface area of 0.24 m$^2$, as a comprimise between efficiency and ensuring that enough leaf material was sampled. These design choices follow established best practices (Černỳ et al., 2019), ensuring that the data collected on leaf litterfall dynamics are reliable and that external factors have minimal influence on LAI estimation.





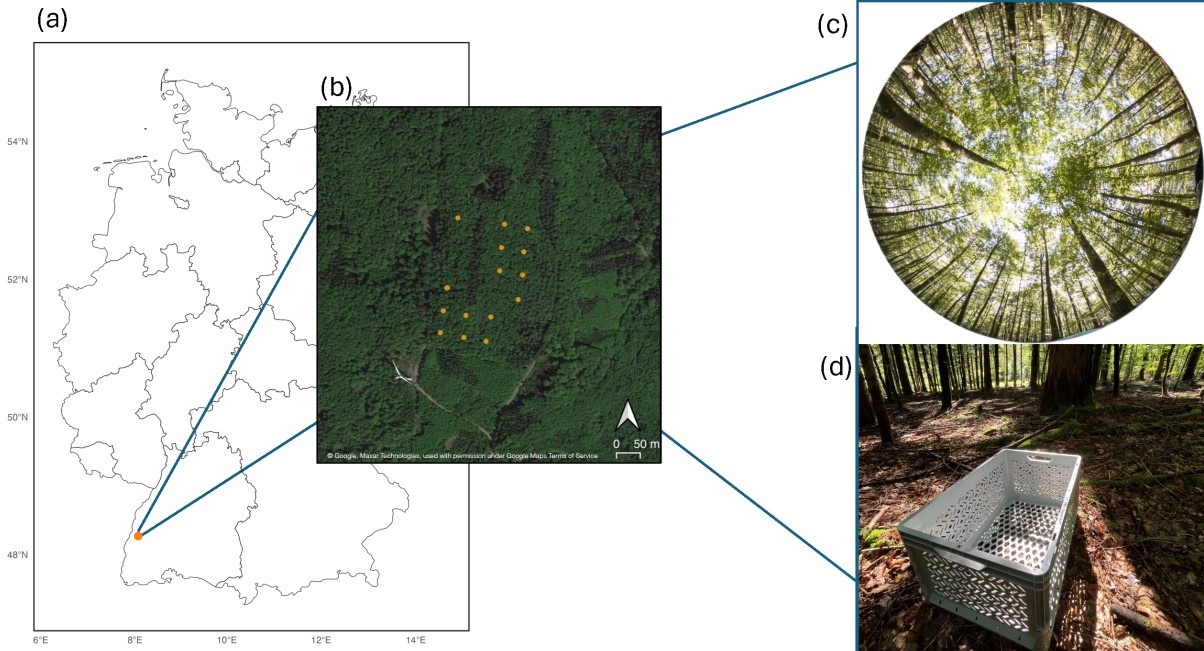

**Figure 1.** (a) Geographical location of the study site in Germany, situated near Ettenheim at the ECOSENSE forest site (center coordinates: 48.2679°N, 7.8783°E). (b) Detailed map of the study area, illustrating the spatial distribution of sample points used for litter trap (LT) and digital hemispherical photography (DHP) overlaid on satellite imagery (© Google, Maxar Technologies, used in accordance with Google Maps/Earth Terms of Service: https://www.google.com/permissions/geoguidelines/). (c) Representative DHP image captured prior to leaf fall. (d) Example of a litter trap installation at a sampling point

For all sample points, LTs were emptied eight times in an approximately bi-weekly setting throughout the leaf fall phase (Figure 2), resulting in a high-resolution time series. To measure the leaf area of each timestemp, and ultimately the LAI, we employed a photograph method. We systematically arranged all collected leaves from LTs at each time step on a 2×1 meter white background. The leaves were carefully pressed flat using a Plexiglass sheet to minimize distortions caused by folds or other deformations that could affect the accuracy of leaf area measurements. Subsequently, photographs were taken from a top-down perspective using a Sony Alpha 7 camera equiped with a 35 mm lens. The images were automatically rectified orthographically using ArUco markers placed in the corner of the white background (Figure A1). The leaves were identified using pixel-wise segmentation, applying grayscale thresholding (threshold value: 240) to generate a binary mask that effectively distinguished leaf regions from the background (Figure A1) (Ghazal et al., 2019). The total leaf area was calculated by summing the detected pixels and converting them to m$^2$. To calculate the total LAI, the leaf area of the leaves inside the LT was divided by the trap area for each time step. These values were then summed over time. For each subsequent time step, the LAI was calculated by subtracting the cumulative LAI of all previous time steps from the total (Figure 2). This method





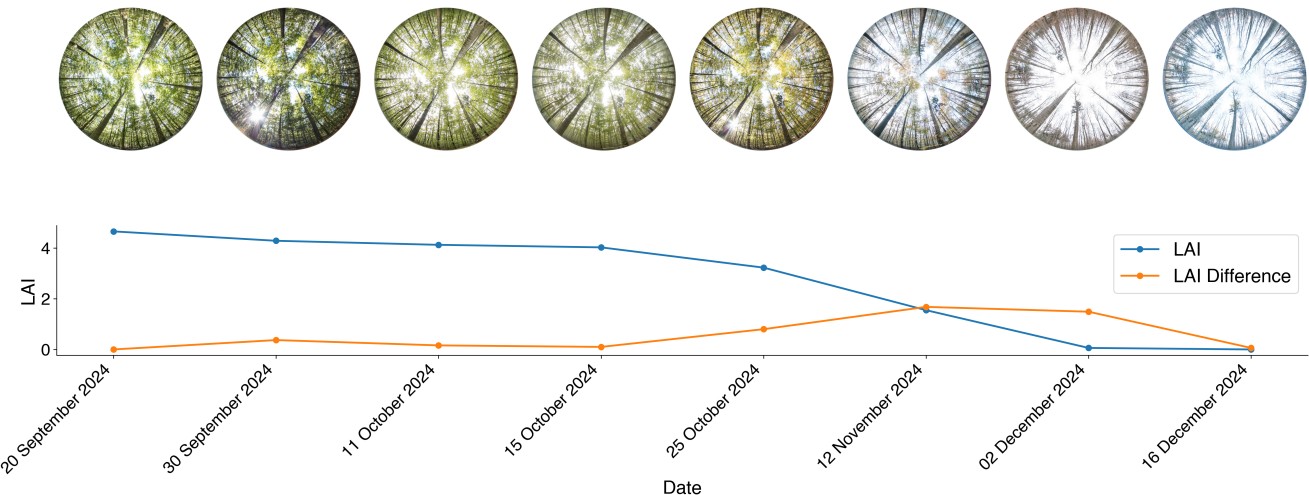

**Figure 2.** Example of digital hemispherical photography (DHP) at a single sample point across all measurement dates. The line plot represent the litter trap (LT) LAI, with the absolute LT LAI values (blue), and the LT LAI differences (orange) for each timestep. The LT LAI values are calculated from the cumulative LAI differences, starting from 0 on 16 December 2024.

provides a more consistent and species-independent alternative to the commonly used LAI approximations using the Specific Leaf Area (SLA), which is based on the dry weight of the leaves and is, hence, influenced by species, seasonal variation, and within tree differences based on the height at which the leaves are collected (Nouvellon et al., 2010; Ishihara and Hiura, 2011; Majasalmi et al., 2013; Eriksson et al., 2005).

### 2.2.1 Digital hemispherical photos

DHPs were acquired using a Canon 700D equipped with a Sigma 4.5 mm lens ensuring a 180° Field of View (FOV). To maintain consistency across all measurements, camera settings were standardized throughout the study. The camera was carefully leveled using a bubble level to ensure precise image alignment across the time steps. Photographs were taken at breast height (1.3 meter) (Schaefer et al., 2015) with a fixed azimuth angle and oriented to the local zenith. One DHP was captured per sample point around noon, coinciding with the emptying of the LTs (Figure 2). The procedure resulted in the acquisition of 120 DHPs.

### 2.3 Data processing

### 2.3.1 DHP analyis

DHP methods estimate the gap fraction based on a binarization of sky and canopy pixels. For this, we used the Otsu method from Otsu et al. (1975), which is effective for canopy images with fish eye lens. This method determines the optimal threshold



for image binarization by analyzing pixel intensity distribution (Tian et al., 2024). It calculates the within-class variance for all possible thresholds and selects the one that minimizes it, ensuring the best separation between foreground and background (Pueschel et al., 2012; Grotti et al., 2020). Light interception, gap fraction and hence LAI estimates highly depend on the leaf angle distribution (Utsugi et al., 2006; Pisek et al., 2013; de Mattos et al., 2020). Therefore, we utilized the *hemispheR* package

(Chianucci and Macek, 2023) in R to process hemispherical canopy images, which enables to estimate and correct for the mean leaf angle. In *hemispheR* package, mean leaf angle is derived by fitting a theoretical ellipsoidal leaf distribution model to observed gap fraction data, optimizing the leaf inclination parameter to best match measured values (Chianucci and Macek, 2023).

The DHP-based LAI retrieval was performed with five zenith rings and eight azimuth segments (Figure A2). Chianucci et al.

(2015) demonstrated that segmenting zenith rings into multiple azimuth sections accounts for non-random foliage distribution and improves estimation of clumping. Additionally, Chianucci and Cutini (2013) found that finer segmentation enhances gap fraction calculations, which is particularly important in dense canopies. While the optimal configuration remains uncertain, this setting represents a practical middle ground for reliable canopy structure assessment.

We evaluated various clumping indices implemented in *hemispheR* package to improve the accuracy of LAI estimation. The

effective LAI (Le) represents a simplified measure of LAI based on the average gap fraction, without accounting for clumping effects. It serves as the baseline for applying clumping indices, whereas actual LAI (L) incorporates clumping at larger scales by averaging the logarithms of gap fractions. The clumping index LX is defined as the ratio of Le to L, providing a basic correction for canopy aggregation. The LXG method, where 'L' stands for Lang, 'X' for Xiang (referencing the original LX method), and 'G' for gap size, combines the finite-length averaging approach with gap size distribution analysis to provide a

more robust clumping correction (Chianucci et al., 2019; Leblanc et al., 2005). LXG1 and LXG2 by Chianucci et al. (2019) improve upon the LX clumping index by integrating ordered weighted gap fraction averaging, which accounts for gap size distribution, reducing sensitivity to spatial scale and canopy density. Unlike LX, which relies on simple averaging the gap fraction, LXG1 assigns greater weight to smaller gaps, while LXG2 emphasizes larger gaps more progressively, leading to better alignment with direct leaf area measurements. This makes LXG1 and LXG2 particularly useful for heterogeneous canopies.

Since DHP primarily measures plant area index (PAI), we subtracted the last measurement of DHP-derived LAI, where only woody material was present, from all sample acquisition dates (Zhu et al., 2018; Fang, 2021; Liu et al., 2015a). This approach allowed us to isolate LAI values without the influence of woody material, ensuring a more accurate assessment of foliage contribution.

## 2.4 Comparing LAI derived from DHP and LT

We compared the LT-derived LAI due DHP-derived LAI estimates obtained with varying zenith angles of the photographs (VZA). In this study, the term VZA refers to the range from 0° to the specified VZA value. Specifically, a VZA of 20° represents the range 0–20°, while a VZA of 30° corresponds to 0–30°, and so forth for all VZA values. This definition applies throughout the study whenever VZA is mentioned (Figure 3). Additionally, we tried the Hinge-angle method which assumes that canopy gaps are more evenly distributed around 57.5°, minimizing the impact of clumping and leaf distribution irregularities (Chen





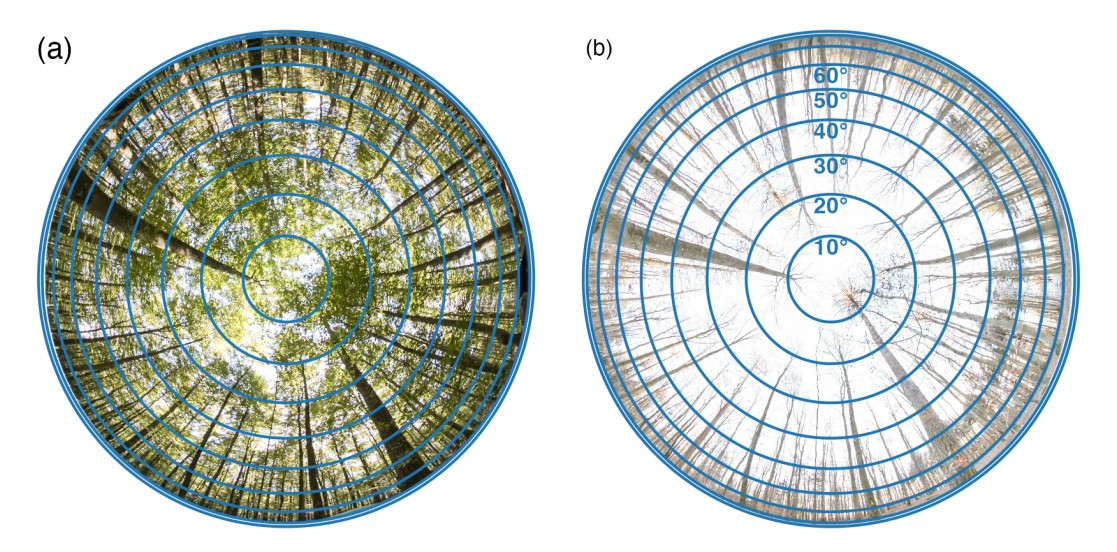

**Figure 3.** Showcase of the area used for LAI calculation in *hemispheR* package at varying view zenith angles (VZA) ranges during leaf on (a) (20 September 2024) and leaf off state (b) (16 December 2024). The VZA ranges always start from 0° to the end VZA shown in (b).

and Black, 1992; Weiss et al., 2004; Wilson, 1963). The Hinge-angle method was implemented in *hemispheR* using a small ring range of 55° to 60°, which limits the calculation of clumping due to the insufficient number of rings and segments for a comprehensive assessment. We compared the LT and DHP-derived LAI using all combinations of clumping methods -including no clumping- and VZA. The agreement for each site and across the time series was assessed using absolute error, slope, and coefficient of determination ($R^2$), derived from a linear regression model.

Furthermore, we evaluated the ability of DHP to represent actual LAI for different leaf fall phases using the best-fitting VZA and clumping indices. To classify the phases we investigated the temporal patterns into three phases: onset, peak, and end of leaf fall, using a breakout calculation with the *segmented* package (Muggeo, 2008) in R (Figure 6). This approach identified breakpoints based on significant changes in LAI values.

### 2.5  Site-specific calibration

We evaluated a site-specific calibration of DHP-based LAI estimates using the LT-derived LAI to evaluate if local biases can be overcome for an efficient and accurate long-term monitoring. We implemented the site-specific calibration using a generalized linear mixed model (GLMM). Since the data were not normally distributed and exhibited left skewness, the GLMM was particularly well-suited for handling these distributional characteristics. Additionally, by including a random intercept, we aimed to minimize the influence of sample point-specific errors and capture the underlying site-specific differences more

accurately. Specifically, we applied a GLMM with an inverse Gaussian distribution and a log link function using the *glmer* function from the *lme4* package (Bates et al., 2015). This approach allowed us to incorporate LAI values adjusted for the



LXG1 clumping index, further refining our site-specific calibration. The following equation describes the generalized linear mixed model (GLMM) used in the analysis:

$$\text{LT LAI} \sim \text{DHP LAI} * \text{leaf fall phase} + (1|\text{sample point}) \tag{1}$$

This formulation accounts for random intercepts by incorporating a random intercept for each individual sample point, recognizing that each sample point may have its own unique random influences (such as local variations in tree density, tree sizes, or growth conditions) while preserving the overall relationship between LT and DHP derived LAI across different leaf fall phase as a fixed effects. The interaction between leaf fall phase and DHP LAI allows for phase-specific slopes, capturing differences in the relationship between DHP and LT LAI across different stages of leaf fall. This model can be applied for similar sites,
particularly in areas with predominantly deciduous tree cover.

All analysis was done in R version 4.2.2 (R Core Team, 2022).

## 3    Results

### 3.1    Spatial footprint of litter traps

We determined the spatial footprint represented by LT using different combinations of VZA and clumping indices. The Hinge
method consistently demonstrated a lower $R^2$ value and a reduced slope (0.3), as well as a higher absolute error (1.73), compared to the optimal combination of the clumping method and VZA. Consequently, it was excluded from further analysis. VZA between 20° and 50° exhibited the lowest absolute error, with a minimum at 20° for LXG1 (0.574), corresponding to 6% of the highest measured LAI for this clumping index. The slope between LT and DHP-based LAI gradually decreased with increasing view zenith angle. The lowest systematic bias was observed between 20° and 30°, where slopes were closest to 1. At 10°, there
was a tendency for overestimation, while for VZA above 30°, LAI estimates increasingly underestimated true values. LXG1 showed the least systematic bias at 20° (slope = 0.942). In contrast, $R^2$ showed a strong increase from 10° to 30°, after which it remained at a high level. The maximum $R^2$ was observed for effective LAI at 60° (0.885), with similar peak values for LXG1 (0.884), LX (0.875), and LXG2 (0.872) (Figure 5)

### 3.2    DHP-derived LAI estimation over time


To assess the ability of DHP to accurately predict temporal LAI dynamics, the mean values from all plots for the LT were compared to the mean values from all plots for DHP using the LXG1 clumping index, which exhibited the closest to 1:1 slope with minimal error (Figure 3). The comparison reveals that the temporal trends of DHP-derived LAI and LT-derived LAI are similar, with the mean values closer together at the end of the time series and an increased spread at the onset of leaf fall.
Moreover, the 5th-95th percentile range is broader at the start of leaf fall with a slight underestimation and becomes narrower towards the end of leaf fall (Figure 5).





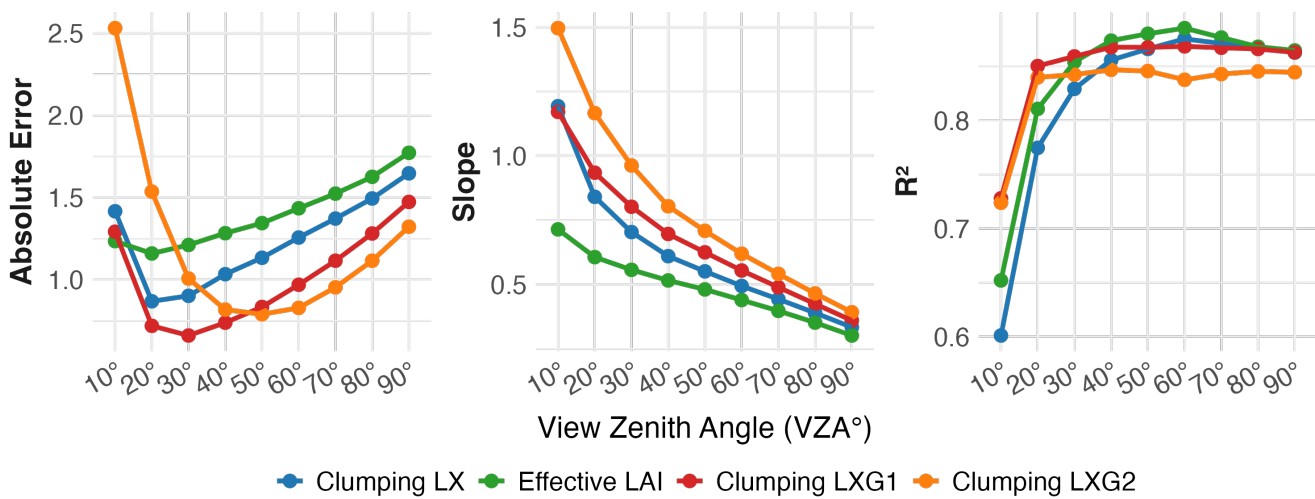

**Figure 4.** Absolute error, slope, and $R^2$ for DHP-derived LAI versus LT-derived LAI across all clumping indices (effective LAI without clumping) and view zenith angles (VZA) from 10° to 90°. The slope is derived from the linear regression between DHP-derived LAI and LT-derived LAI, indicating how the DHP estimates compare to the LT reference. The absolute error is calculated as the mean residual from the 1:1 line, representing the deviation of DHP estimates from the ideal agreement with LT-derived LAI. $R^2$ quantifies the strength of this relationship. Data points represent computed values at each VZA, with lines connecting them to illustrate trends across different clumping indices.

In order to further investigate the LAI estimation over time, we set up three phases (onset, peak and end) of the leaf fall to assess those phases separately (Figure 5). We analyzed the differences of leaf fall phases based on the LXG1 LAI (with clumping). The analysis shows almost no systematic over- or underestimation (Figure 6).

## 3.3 Site-specific calibration

We selected the LXG1 clumping index for the site specific calibration because it produced the lowest absolute error while maintaining almost no systematic over- or underestimation with high correlation in comparison to LXG2 which had higher systematic over-/ underestimation with similar correlation. At a VZA of 20°, the site specific calibration reduced the absolute error by 52%, from 0.574 to 0.275, demonstrating a substantial improvement in accuracy. The systematic over- underestimation and correlation stayed at a similar level with a slope increase of 0.02 and an increase in R2 from 0.88 to 0.97 (Figure 6).





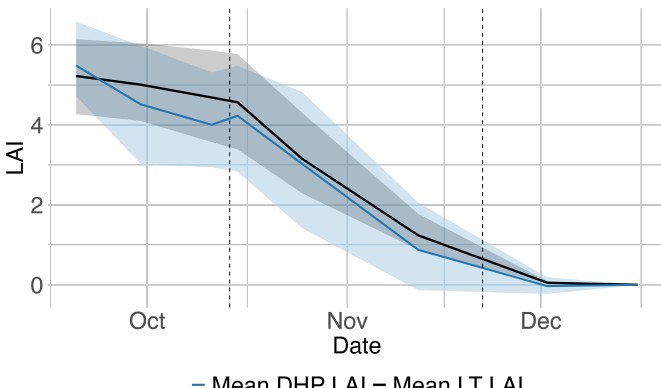

**Figure 5.** Comparison of LT LAI and DHP LAI (estimated with VZA 20° and clumping index LXG1) over time for all sample points, showing the mean values and the 5th to 95th percentile for each time point. The solid black line represents the mean of actual LAI for all sample points, while the blue line shows the mean of DHP LAI for all sample points. The shaded regions indicate the 5th-95th percentile range for both LT LAI and DHP LAI, highlighting the variation and the spread of the data over time. Dashed vertical lines represent breakpoints identified in the segmented regression analysis, marking significant shifts in LAI trends.

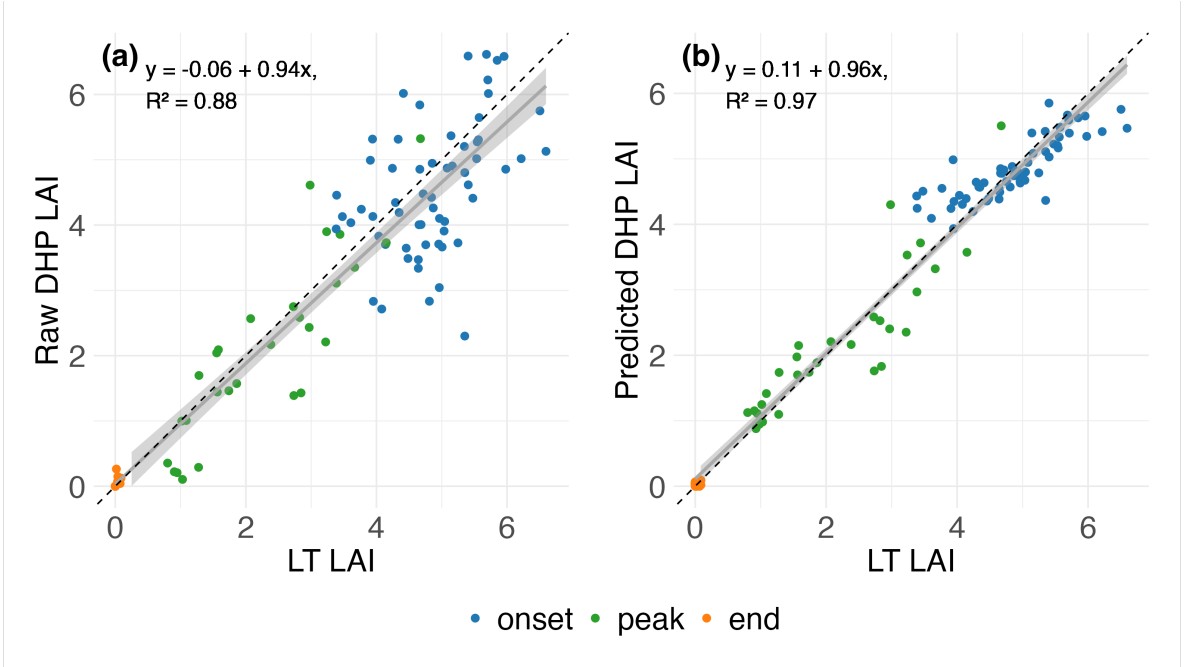

**Figure 6.** (a) DHP-derived LAI using the clumping Index LXG1 for a View Zenith Angle of 20° against LT LAI. The sample points are colored according to the phase of leaf fall. The breakpoints for the leaf fall phases were determined using a breakpoint analysis and are shown in Figure 5. (b) predicted LAI of the GLMM against the LT LAI, illustrating the improved accuracy of the site-specific calibration.





## 4 Discussion

### 4.1 Spatial footprint and influence of zenith angle

Our findings demonstrate that the spatial footprint of LTs corresponds well with specific VZA ranges, enhancing our under-
standing of LAI estimation using DHP. By correlating LT-derived LAI with DHP-derived LAI across multiple VZA ranges
(0°–10° to 0°–90° in 10° increments), we identified an optimal VZA range (20°) for DHP analysis.

The Hinge Angle method, which has shown strong performance in other studies (Liu et al., 2021), did not align as well with
LT-derived LAI in our dataset and was therefore less suitable for our analysis, although no clumping index was applied with the
implementation using *HemispheR*. Instead, clumping indices proved crucial in improving DHP accuracy, particularly at lower
VZA values, where they helped reduce LAI underestimation compared to effective LAI without clumping corrections (Figure
3). Among the tested clumping indices, LXG1 exhibited the best performance, as it is particularly effective in accounting for
smaller canopy gaps and performing well in dense broadleaf forests (Chianucci et al., 2019; Chianucci and Macek, 2023).

We observed that the correlation between DHP-derived and LT-derived LAI was highest at lower VZA values (< 30°) and
decreased with increasing VZA. The spatial footprint of LTs was best represented at a VZA of 20°, supporting previous
recommendations to use a VZA range of 20°– 40° for DHP-based LAI estimation in temperate forests (Lee et al., 2023).
Additionally, this raised the question of whether the 20° VZA truly represents the spatial footprint of LT or if the decreasing
correlation at higher VZA values is primarily due to greater epistemic errors of the DHP method. Given that the best VZA
range was previously assumed to be 20°–40°, and the correlation remained strongest at 20°, we infer that epistemic errors play
a minor role. Furthermore, it should be noted that both the slope decreased and error had already increased significantly from
20° to the 40° VZA, which is still assumed to be an appropriate VZA range for DHP estimations. Thus, we consider 20° VZA
to be a reasonable setting for accurate LAI estimations.

Contrasting to our results of an ideal VZA range of 0-20°, Liu et al. (2015b) recommended a VZA range of 30-60° VZA
range. At 30-60° VZA, we observe an underestimation of LAI, which suggests that further investigation into the impact of VZA
range on estimated LAI from DHP would be valuable. Our finding that LAI underestimation increased with higher VZA values
is consistent with previous research attributing this effect to larger canopy gaps and overestimated openness (Geng et al., 2021).
Also note that a small VZA range also offers to resolve spatial LAI variation within small spatial gradients. Such focused areas
are well-suited for validating LAI products from high-resolutions satellite data, which often have spatial resolutions around
10-30 meters (e.g. Sentinel-2 or Landsat data).

As shown by our results, the choice of clumping index remains critical, as effective LAI values without clumping corrections
showed significant underestimation and higher absolute error. Here, the best performing index was the LXG1. This clumping
index is similar to the CLX index (Liu et al., 2015b), which incorporates gap fraction averaging as a core principle and has
shown good results for seasonality. The key difference between the two indices is that LXG1 focuses on refining weighting
functions for segment-based calculations, while CLX emphasizes gap fraction averaging.

Overall, our findings emphasize the importance of carefully selecting VZA ranges and incorporating canopy structure cor-
rections in DHP-based studies. While VZA is often underemphasized in DHP research, our results demonstrate that it has





a substantial impact on absolute error and systematic bias. Future studies should prioritize VZA optimization and clumping index selection to enhance the accuracy of DHP-derived LAI estimates.

## 4.2 Temporal representation of DHP-derived LAI

Our results demonstrate that DHP with LXG1 effectively captured seasonal LAI trends, accurately reflecting variations throughout the growing season. Discrepancies decreased throughout the leaf fall period, ultimately resulting in no discrepancies at the
final timestep due to the imposed value of 0 for both DHP and LT LAI. This reduction results from subtracting woody material from the PAI like previously done from Zhu et al. (2018), ensuring that only the LAI is analyzed. The fraction of woody material is known to be the main cause of error in deciduous stands shown Zhu et al. (2018). As we substracted the LAI from the final time step after the leaf fall, showing only woody material, our results are not affected by systematic overestimation. In addition, we observe higher variability in the onset phase due to the increased structural complexity of fully developed canopies,
leading to greater variability in LAI estimates due to the heterogeneous light environment and mutual shading effects, which cannot be fully accounted for by clumping alone (Liu et al., 2015b).

## 4.3 Site-specific calibration using a generalized linear mixed model

To address local variability in DHP-derived LAI estimates, we implemented a site-specific calibration using a generalized linear mixed model (GLMM). This model incorporated the LXG1-adjusted LAI as a fixed effect, with individual sample
points included as random intercepts to account for spatial heterogeneity. The LXG1 clumping index was selected based on its superior performance in minimizing systematic error across all VZA configurations (Figure 3).

Importantly, we integrated the leaf fall phase as an interaction term, allowing the model to adjust slope parameters depending on the temporal dynamics of leaf fall. This phase-specific correction significantly improved model fit, particularly during the early stages of leaf fall where variability in canopy structure is highest. At a view zenith angle (VZA) of 20°, the calibrated
model reduced the absolute error by 52%, with the largest improvements observed during the onset phase, particularly for LAI values between 4 and 6 (Figure 6).

These results underscore the value of combining DHP-derived LAI with LT reference data in a mixed-effects modeling framework. By explicitly modeling spatial and temporal sources of variation, the approach enhances the robustness and accuracy of LAI estimation across different forest conditions. This site-specific model can be extended to similar temperate
deciduous forests, provided that data collection includes temporal labeling of leaf fall phase. Notably, since the leaf fall phases align with distinct LAI value ranges, phase classification could also be approximated directly from observed LAI trends in future applications.

## 4.4 Challenges and future perspectives

DHP-derived LAI is influenced by multiple factors that can introduce systematic biases. Variability in image quality, canopy
cover, tree size and lighting conditions affect the segmentation of canopy and sky, impacting the accuracy of LAI estimation



(Chianucci, 2020). This variability in image quality, canopy cover, tree size, and lighting conditions further underscores the robustness of the DHP-derived LAI estimation. Despite these variations, the strong correlation between the DHP and LT LAI data highlights the reliability and consistency of the DHP method across diverse conditions within the study site.

The parameterization of azimuthal segments and zenith rings influences clumping index calculations, as these parameters
define foliage distribution patterns and serve as the basis for clumping correction (Chianucci and Macek, 2023). However, determining optimal parameterization remains challenging. In later leaf fall stages, a reduced number of azimuth segments may be sufficient, as increased canopy openness diminishes the need for detailed segmentation (Liu et al., 2015a).

LT data are primarily suitable for deciduous forest stands, as they accurately capture seasonal leaf shedding but do not effectively represent evergreen canopies (Jonckheere et al., 2004). In this study, sample points containing conifer trees were
excluded due to the lack of validation data for DHP. While DHP methods are highly reliable for deciduous forests, they can also be adapted for coniferous stands. However, an alternative validation approach is required, as LT-based LAI estimation is unsuitable for conifers, given that litter traps primarily capture fallen foliage and do not account for retained needles.

Future research should explore allometric approaches for validating DHP-derived LAI in evergreen forests. One promising method involves deriving LAI for individual branches through direct, destructive measurement, and subsequently upscaling
these estimates to the entire canopy by counting branches per tree (Flynn et al., 2023; Badea, 2011). This upscaling can be assisted by structural data from Terrestrial Laser Scanning (TLS) or photogrammetry, which allows for the detailed quantification of branching architecture. Such an approach provides a non-destructive, yet scalable reference for validating DHP in coniferous stands and could greatly enhance LAI estimation in evergreen-dominated ecosystems.

## 5 Conclusion

This study demonstrates that DHP-based LAI when adjusted with the LXG1 clumping Index (Chianucci et al., 2019), closely aligns with LT-based LAI in temperate deciduous forest canopies. The strongest agreement between the two methods was observed at 20° VZA ($R^2$ = 0.884), indicating that LT-derived LAI captures the variability of LAI within relatively small spatial footprint. While showing strong agreement for all VZA values above 20° the data showed stronger underestimation and error for greater VZA ranges. Thus we can see that VZA has a major role in the underestimation of LAI and therefore needs to be
accounted more in future studies. DHP-based LAI estimation over time showed that there is a tendency to underestimate LT LAI in the beginning of leaf fall, with higher variability in the sample points, which is influenced by the structural complexity of a fully developed canopy. To address these limitations, we applied a GLMM that incorporated LXG1-adjusted LAI, which significantly improved site-specific precision in DHP-based LAI estimation. Overall, this study provides a robust framework for improving DHP-based LAI assessments, reducing both measurement error and the need for labor-intensive LT collection.
However, DHP calibration remains a challenge for coniferous forests, where needle retention and canopy structure introduce additional complexities. Future research should focus on developing and validating calibration approaches for DHP in coniferous stands. Expanding these methodologies will enhance the applicability of DHP-based LAI estimation across diverse forest ecosystems.



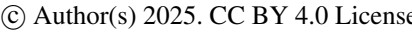

**Code and data availability**

The code used in this study is available in the public GitHub repository: https://github.com/lotzsi/LittervsLens.

**Author contributions**

SL, TK and TJ contributed to the conceptulization. SL, NK and TK prepared the first draft. SL, TJ, SS and TK contributed to the data acquisition. SL, TK, JF and TJ contributed to the methodology. SL and AG contributed to data analysis. NK and TKcontributed to supervision. SL, NK and TK contributed to the original draft preperation. SL contributed to visualization and

formal analysis. TK contributed to funding acquisition and resource acquisition. All authors contributed to review and editing.

**Competing interests**

The contact author has declared that none of the authors has any competing interests.

**Acknowledgments**

The study was funded by the Eva Mayr-Stihl Foundation, »XR Future Forests Lab« at the Faculty of Environment and Natural
Resources, University of Freiburg. The authors are grateful for data provided by Matthias Gassilloud and field campaign support by Markus Quinten.

**Financial support**

This research has been supported by the German Research Foundation (DFG) under the collaborative Research Centre ECOSENSE (SFB1537).

**Appendix A: Appendix A**

**A1   LT Leaf Area**

**A2   hemispheR**

**A3   canopy cover and mean height**



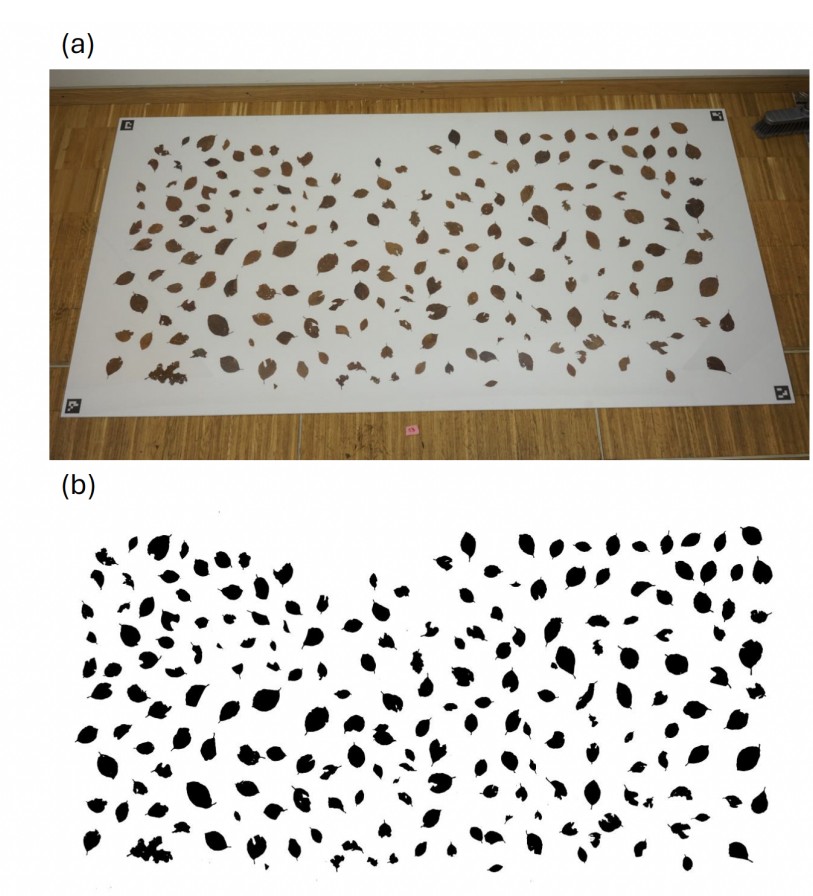

**Figure A1.** Leaf area measurement process from litter traps. (a) shows the original image of the collected leaves, spread out on a 2×1 meter background and pressed flat with a Plexiglass sheet to reduce deformations. (b) demonstrates the rectified image with the binary mask of the leaves, generated by applying grayscale thresholding (240) for pixel-wise segmentation. The mask separates the leaf regions from the background.

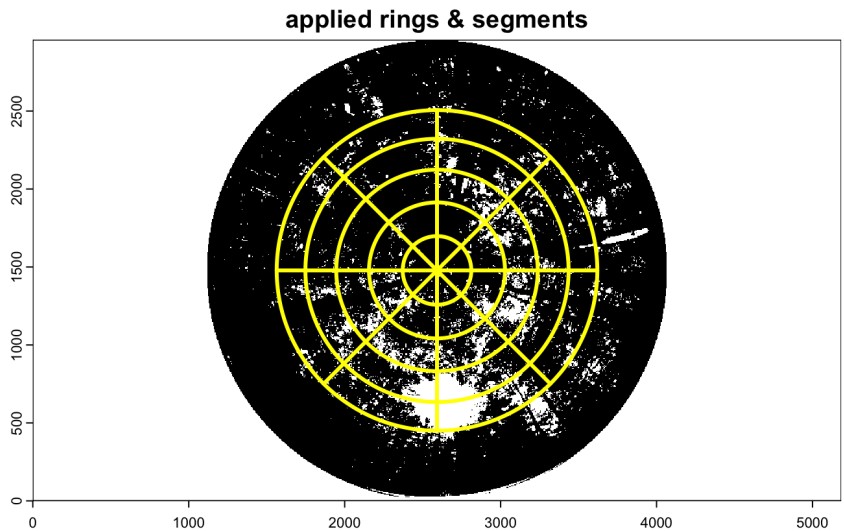

**Figure A2.** This is an example illustrating the different zenith rings and azimuth segments using one of the VZA ranges used. In this case it is 0°–60°.



| Sample Point | Height 5m (m) | Cover 5m (%) | Height 10m (m) | Cover 10m (%) |
|---|---|---|---|---|
| LT11 | 25 | 97.7 | 24 | 97.0 |
| LT13 | 25 | 96.2 | 24 | 96.5 |
| LT14 | 26 | 98.4 | 25 | 97.6 |
| LT23 | 27 | 93.6 | 26 | 96.5 |
| LT24 | 24 | 98.8 | 25 | 98.7 |
| LT33 | 27 | 97.9 | 27 | 98.4 |
| LT34 | 27 | 98.5 | 27 | 98.6 |
| LT41 | 23 | 98.8 | 23 | 98.7 |
| LT44 | 25 | 98.6 | 24 | 97.7 |
| LT51 | 25 | 98.3 | 25 | 98.0 |
| LT52 | 27 | 97.6 | 28 | 97.8 |
| LT53 | 26 | 98.2 | 26 | 98.2 |
| LT62 | 22 | 96.8 | 23 | 96.7 |
| LT61 | 24 | 96.4 | 24 | 94.9 |
| LT63 | 26 | 98.2 | 26 | 98.4 |
| **Average** | **25** | **97.6** | **25** | **97.6** |

**Table A1.** Sample Point Data for Mean Height and Canopy Cover. These statistics were derived from a drone-based LiDAR point cloud (DJI Matrice equiped with an L2 flown at 60 m height). The statistics were derived from a canopy height model and buffers at 5m and 10m radius around the sample points. The last row displays the average of all sample points for the corresponding column.



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
