# Peer review of "Litter vs. Lens: Evaluating LAI from Litter Traps and Hemispherical Photos Across View Zenith Angles and Leaf Fall Phases"

_EGUsphere, 2025_

## Author Comment (AC3)

Simon Lotz
Faculty for Environmental and Natural
Resources
Sensor-based Geoinformatics lab
Tennenbacherstr. 4, 79116, Freiburg,
Germany
simon.lotz@posteo.de

To Biogeoscience EGUsphere Journal

Oct 17th 2025

**egusphere-2025-1496- "Litter vs. Lens: Evaluating LAI from Litter Traps and Hemispherical Photos Across View Zenith Angles and Leaf Fall Phases"**

Dear handling associate editor Prof. Dr. Paul Stoy, dear reviewers

We would like to thank all three reviewers for their favorable and constructive comments that allowed us to improve the quality of the manuscript.

We carefully considered each point and revised the text accordingly. Below, we provide a detailed, point-by-point response to all comments. All changes made in the revised version are highlighted in blue font . The main improvements of the manuscript include:

- New diagnostic analyses to address reviewer RC1's request for decomposing DHP-derived LAI components—examining the variation of gap fraction (GF), clumping index (CI), and leaf projection function (G(θ)) across view zenith angles (VZA).
- Clarification of data collection procedures for hemispherical photography, including exposure control, sky condition and camera setting, and an explicit discussion of associated limitation
- Enhanced theoretical background in the introduction, including the Beer-Lambert law, the
  distinction between effective and true LAI, and the role of clumping index in optical LAI
  estimation.
- Refined figures and captions, with improved font sizes, color-blind–friendly palettes, and clearer labeling of equations and panels.

We hope the revised manuscript meets the journal's requirements and look forward to your decision.

Kind regards, Simon Lotz (on behalf of the co-authors)

**egusphere-2025-1496- "Litter vs. Lens: Evaluating LAI from Litter Traps and Hemispherical Photos Across View Zenith Angles and Leaf Fall Phases"**

Find below a point-by-point response. Within the track-change version of the revised manuscript, the changes are highlighted with blue font.

**Replies to comments from reviewer #RC1 Francesco Chianucci**

The changes are highlighted in a track-change version with blue font.

1) "data collection. Fisheye photos are strongly influenced by sky condition and camera exposure. The authors collected data in direct sky conditions and likely automatic exposure. Both options greatly influence gap fraction retrieval, with generally increase the underestimation of LAI from DHP. This is apparent in the sample images in Fig2-Fig3 where the sky is likely saturating (clipping) and this can largely overestimate gap fraction (and thus underestimate LAI). Additionally, gamma correction is another factor required to provide linear relationship between image brightness and radiance.

There is a wide range of literature confirming this, and therefore the accuracy of the method can be biased compared to litter traps. This should be acknowledged and authors could try to find some solution to this issue. For example, if they collected RAW imagery, they can try to analyse raw images using the bRaw R package, https://www.biorxiv.org/content/10.1101/2022.10.25.513518v1, which resemble a method by Macfarlane et al. 2014, to linearly scale 12 to 8 bit raw blue imagery to reduce DHP sensitivity to camera exposure."

**Author response:**

We thank the reviewer for the constructive comments and suggestions. We acknowledge that fisheye photos are sensitive to sky conditions and exposure settings. During data collection, most images were taken at f/5.6 and 1/100 s. We always checked the pictures by experts in the field. When overexposure occurred, we increased the aperture value to minimize overexposure.

We agree that varying sky conditions may affect gap fraction estimates. We could not standardize these conditions due to logistic constraints. Unfortunately, raw images are not available for all dates, preventing full re-analysis with the suggested method. Nevertheless, results remained consistent across conditions, indicating limited bias. We will discuss this limitation and cite the suggested references in the revised manuscript as follows in L 272.

"Fisheye imagery is particularly sensitive to sky conditions and exposure settings, which can alter gap fraction retrieval and thereby bias LAI estimates. Most images in this study were acquired under diffuse or moderately bright conditions using an aperture-priority mode (f/5.6, –1 EV compensation, 1/100 s exposure), with visual inspection by experienced operators to minimize overexposure. Nevertheless, occasional variations in illumination and exposure could have influenced canopy—sky classification, particularly under high-contrast sky conditions. While we attempted to maintain consistent settings, logistical constraints prevented complete standardization across sampling dates. Despite these potential limitations, the consistency of DHP–LT correlations across measurement dates suggests that exposure variability introduced only minor bias in our dataset. Future applications should consider capturing and processing

RAW fisheye images to ensure linear radiometric scaling and reduce sensitivity to camera exposure. "

2) "The DHP invert LAI without requiring information on extinction coefficient by integrating multiple gap fraction (GF) measurements over the full zenith range, as the integral of G(theta) is 0.5 over 0-90°, as per the Miller 1967 theorem. By reducing the zenith angle view, the LAI inversion is not straightforward. Therefore, the higher correlation with LT at smaller VZA is in my view the results of higher clumping correction, as the inner rings could have more variability, which reduced the underestimation of DHP. Authors should de-structure their LAI component obtained from DHP, namely explore the variation in GF, CI and G(theta) (here using LAI from LT) across VZA, to explore how these three attributes varied with VZA. This would shed some light on the different performance of canopy units like GF, G and CI to the resulting LAI estimation from DHP."

**Author response:**

We thank the reviewer for this valuable suggestion and performed an analysis on this. The reviewer is right, that indeed there are higher clumping corrections at smaller VZA. We de-structured this and will add the following figure as figure 5:

**Figure 5.** Variation of gap fraction (GF), clumping indices (CI; LX, LXG1, LXG2), and leaf projection function ( $G(\theta)$ ) derived from DHP across view zenith angles (VZA), averaged over all sampling points and dates. Error bars indicate the standard error across sites.

To explore which of the various factors determines the LAI retrieval accuracy, we compared the effect of clumping index, gap fraction, G(theta) and viewing zenith angle. To more explicitly represent the footprint size of the DHP imagery, we derived the footprint size in form of the radius for a given VZA.

 $r(\theta)$ =(Height*canopy*-Height*camera*)tan( $\theta$ )

And explored the influence of all parameters on the error of LT-DHPLAI with a random forest model and the following variable importance:

randomForest(abs\_error ~ GF\_scaled + LXG1\_scaled + G\_theta\_scaled + radius\_scaled)

**Variable Importance**

| GF_scaled      | 39.20396 |
|----------------|----------|
| LXG1_scaled    | 40.25627 |
| G_theta_scaled | 37.95073 |
| radius_scaled  | 52.96107 |

As shown by these results, it appears that all factors do influence the LAI retrieval. We would like to thank the reviewer for highlighting that the GF, CI, G(theta) may also play a considerable role, something which we previously did not consider sufficiently. In the revised manuscript we will therefore include these results and discuss that the improved retrieval at smaller VZA is a compounded effect of a) more spatially explicit correspondence of litter traps and DHP photographs as well as b) and improved compensation of the clumping index.

We will implement how we analyzed the canopy units in Line 155 in the Methods section. The results will be implemented in section 3.1 showing the influencing factors on LAI retrieval including spatial footprint and canopy units.

**3) "Direct, semi-direct vs indirect methods**

LT is typically considered a semi-direct method (Bréda 2003), in that the leaf area is directly measured, as for the allometry methods. No "fully" direct methods exist in forestry. Basically, the distinction is between those based on direct measurement, or contact-approaches (Jonckheere et al. 2004) and those based on optical theory. With "Indirect methods " the literature consider all the methods based on optical theory, and particularly the Beer-Lambert law. Indirect methods in the field are based on both passive and active sensors."

**Author response:**

We thank the reviewer for spotting this inconsistency. Indeed, we measure leaf area directly but only from a sample, for which the result is then extrapolated to a given area. We will revise the introduction as follows:

"One direct A semi-direct method is the litter trap (LT) method, which involves deploying containers of known ground area to collect fallen leaves over time \citep{vcerny2019leaf}. The ratio of the total leaf area accumulated within the trap to its ground area provides an estimate of LAI. LT is, thus, considered a semi-direct method because it relies on actual leaf material but is not fully exhaustive \citep{breda2003ground, jonckheereReviewMethodsSitu2004}."

4) "Importantly, authors should explain the difference between transmittance based and gap fraction based methods. The former requires measuring above and below (absolute or relative) light, and then calculate light transmittance accordingly. This comprises PAR ceptometers . Gap fraction based methods include LAI-2000, which estimate gap as the ratio of above to below canopy relative radiation, and DHP, which calculate gap fraction as the ratio of classified pixels gaps over total image pixels. With some assumption (e.g., leaves are homogeneous turbid medium) gap fraction and light transmittance equals, and then the Beer Lambert law is used on all these methods."

**Author response:**

We thank the reviewer for pointing out the missing explanation of the differences between transmittance-based and gap-fraction-based methods. We fully agree and have clarified this in the revised manuscript as follows:

"Optical LAI estimation methods can be broadly grouped into transmittance-based and gap-fraction—based approaches. Transmittance-based methods, such as PAR ceptometers, quantify the attenuation of photosynthetically active radiation (PAR) by measuring above- and below-canopy light fluxes. LAI is then derived using the Beer—Lambert law, relating light transmittance to leaf area through an extinction coefficient. Gap-fraction—based methods, such as LAI-2000/LAI-2200 or digital hemispherical photography (DHP), instead measure the proportion of visible sky at different zenith angles and invert LAI geometrically via the same theoretical relationship. Under the assumption of a homogeneous, turbid canopy, light transmittance and gap fraction are equivalent, and both methods thus rely on the Beer—Lambert law but differ in the variable observed, that is energy flux versus canopy openness."

5) "Finally, I would add in the intro the well-established Beer-law formula and explain differences between effective to true LAI in optical measurements. This will help introducing the importance of clumping index, its multi-scale spatial nature, and why different methods provide different clumping corrections, and finally, why effective LAI is the first step to know the true LAI, also compared with those from direct methods (which by definition ignore clumping)"

**Author response:**

We thank the reviewer for this valuable suggestion. We will revise the Introduction to include the Beer–Lambert law formulation and a brief explanation of the distinction between effective and true LAI in optical measurements. This addition helps to show the role of the clumping index, its spatial-scale dependence, and its importance in bridging optical and direct LAI estimates. We will revise the text as follow in L40 of introduction:

"The relative difference in PAR transmission is used to estimate LAI via the Beer-Lambert Law, which relates light attenuation to leaf area (Saitoh et al., 2012, see equation 1). However, PAR-based methods assume that plant canopies are a homogenous medium ( $\Omega$  = 1), if this is not corrected using post-hoc gap fraction measurements. Without a clumping correction LAI will thus be underestimated. Thus, in complex canopies, the accuracy of PAR-based methods is constrained due to light scattering, foliage clumping, and saturation at high LAI values (Yao et al., 2016). Accordingly PAR-based methods were found to be inadequate for forests due to the complexity of canopy structure, including branches and stems, leaf clumping and leaf angle variation (Bréda, 2003; Chen and Black, 1992; Geng et al., 2021).

**$T = exp[-G(\theta)\Omega L/cos\theta]$ (1)**

Another indirect method for estimating LAI is using digital hemispherical photography (DHP), which captures wide-angle images of the canopy from beneath using a fisheye lens (Chianucci and Cutini, 2013). Such images can be taken at various sample points within the study area and provide detailed information on canopy gaps, leaf density, and stem distribution. Additionally, DHP allows to directly consider canopy complexity, including clumping effects (compare  $\Omega$  in equation 1)"

6) "L33 the litter traps is also labor intensive cause the litter should be separated, and the leaf component dried in forced-air stove, to determine the dry mass, which is combined with SLA"

**Author response:**

We thank the reviewer for this valuable comment. We agree that litter-trap methods are highly labor-intensive, as they typically require separation, drying, and specific leaf area (SLA) determination. However, in our case we didn't perform SLA-based measurements. Instead, we estimated leaf area optically from photographs. See also comment 9.

7) "L 60 here an example of continuous DHP measurements https://www.sciencedirect.com/science/article/pii/S0168192320300460 "

**Author response:**

Thanks for pointing that out. You are totally right, we will cite the study that you mentioned in L60.

**8) L90. Timestemp/timestep -> sampling period**

**Author response:**

Thanks for the recommendation we will change all timestemp/timesteps to sampling period.

"To measure the leaf area of each sampling period timestemp, and ultimately the LAI, we employed a photograph method."

9) "L90-120 From the best I can understand this approach foresee the indirect (i.e. scanning) measurements of all leaves in the traps. It is noticeably more time consuming

than measuring a sample of leaves to determine SLA and then multiply with the total biomass. Authors should acknowledge this. Additionally, it is the very first time I see a litter trap laying at the floor - these baskets are typically set at 1 m height to reduce seed predation - how these traps be impacted by fauna, which can influence the leaf litter inside?

**Author response:**

We appreciate the reviewer's insightful comments. We agree that scanning all leaves collected from litter traps is more time-consuming compared to estimating LAI using specific leaf area (SLA) measurements and total biomass. However, SLA can vary substantially within and between species, and even across the season, which can introduce additional uncertainty when upscaling from a limited subsample. By directly scanning all leaves, we minimized this source of variability and obtained more consistent and representative estimates of leaf area.

Regarding trap placement, logistical constraints and the absence of significant fauna activity at our study site led us to position the litter traps directly on the ground surface. To minimize potential soil–fauna interactions, a fine net was placed beneath each trap to prevent interference from soil organisms (e.g., earthworms). No evidence of seed predation or litter removal was observed during the collection period.

Finally, leaves were analyzed in fresh condition without drying to avoid shrinkage, as the optical scanning method used for leaf area determination is sensitive to leaf deformation. Given the frequent sampling intervals, we assume negligible leaf decay during the measurement period.

We will clarify that a net was placed inside the boxes to avoid fauna interactions: L97:

"Compared to conventional approaches that determine specific leaf area (SLA) from subsamples and scale to total biomass, scanning all leaves is considerably more time-consuming. However, it avoids uncertainties arising from species-specific or seasonal variation in SLA and provides a consistent, sample-based estimate of leaf area for each collection date."

**L100:**

"All litter traps were placed directly on the ground surface due to logistical constraints and the absence of significant fauna activity at the site. To minimize potential soil—fauna interference, a fine mesh net was placed beneath each trap, preventing access by soil invertebrates such as earthworms. No evidence of litter removal or seed predation was observed during the study period."

**Replies to comments from reviewer #RC2 Anonymous Referee**

The changes are highlighted in a track-change version with blue font.

1) "Check wording and usage throughout the manuscript. One example is the verb in 'Uncertainties for individual litter traps attributed to varying site conditions, such as tree stem density or canopy coverage.' Honestly some of the text reads like it was written or checked by a LLM including 'As leaves constitute the active surfaces for these mass and energy exchanges, the leaf area index (LAI) emerges as a fundamental biophysical parameter - quantifying canopy structural complexity through its definition as half of the total intercepting area per unit ground surface area'. An additional careful round of editing would be beneficial."

**Author response:**

We thank the reviewer for this careful reading and constructive feedback. We confirm that no LLM was used in preparing the manuscript. However, we fully acknowledge that some sentences, particularly in the Introduction, are overly long and could be improved for clarity and readability In the revised version, we will thoroughly review the wording and sentence structure throughout the manuscript to ensure a more concise and natural flow. We appreciate the reviewer's observation, which will help us improve the overall quality of the text.

**2) "On line 31 there are more fundamental references for litter traps. Somewhat disagree that litter traps can't be used for evergreen species, one just has to correct for leaf geometry."**

**Author response:**

We appreciate the reviewer's comment and agree that litter traps can indeed be used for evergreen species in general. Our intention was to state that litter traps cannot be directly used to derive LAI for evergreen species, since their leaves are not shed synchronously within a given measurement period, making it difficult to capture a complete annual leaf turnover. We will revise the text accordingly to clarify this distinction and include additional fundamental references on litter trap methodology.

"One semi-direct method is the litter trap (LT) approach, which collects fallen leaves in containers of known ground area to estimate LAI as the ratio of leaf area to trap area (Bréda, 2003; Jonckheere et al., 2004; Černý et al., 2019). While applicable to both deciduous and evergreen stands, its use for annual LAI in evergreens is limited because leaf shedding occurs continuously rather than synchronously (Chen et al., 1997)."

**3) "40: corrections for clumping can be applied although I guess this is still correct in that the fundamental equation is for a homogeneous medium"**

**Author response:**

We thank the reviewer for this valuable comment. We agree that PAR-based (transmittance) methods can in theory include a clumping correction term  $(\Omega)$  within the Beer–Lambert framework (Nilson, 1971; Chen & Black, 1992), Ω cannot be derived directly from the integrated PAR signal. Consequently, such corrections are rarely implemented in practice. Only a few empirical approaches, such as the stand-specific correction proposed by Lopes et al. (2014), attempt to address this limitation.

**4) "From figure 1d a bit surprised that the litter traps weren't suspended above the ground to help dry material and prevent decay"**

**Author response:**

We thank the reviewer for this observation. We aimed to collect fresh material to minimize shrinkage, since leaf area was determined using an optical method. Samples were collected regularly, assuming no decay of the material.

We referred to this in comment 9) of reviewer RC1 as well. We will clarify this in the revised section as follows:

**L100:**

"All litter traps were placed directly on the ground surface due to logistical constraints and the absence of significant fauna activity at the site. To minimize potential soil-fauna interference, a fine mesh net was placed beneath each trap, preventing access by soil invertebrates such as earthworms. No evidence of litter removal or seed predation was observed during the study period."

**5) "Font sizes in Figure 2 are very small. It's not entirely clear to me what LAI difference means as described."**

**Author response:**

We thank the reviewer for this helpful comment. The "LAI difference" represents the rate of change (slope) of LAI between consecutive measurement dates. We will update the figure caption and description in the text to clarify this and will increase the font sizes for better readability.

**6) "Equation 1: don't use the star for multiplication in formal mathematical equations. Use the multiplication symbol."**

**Author response:**

We thank the reviewer for this valuable suggestion. We will change it to the actual multiplication symbol.

**7) "Figure 4: I question if this color scheme is adequate for people with red-green colorblindness. Different colors should be chosen."**

**Author response:**

We thank the reviewer for pointing this out. We implemented your suggestion and chose a color Scheme that is adequate for colorblindness people using the viridis package.

8) "As noted in other reviews, avoiding sunny conditions is key to not wash out canopy photos. What steps were taken to ensure consistent sky conditions either near dusk or with uniform cloudiness?"

**Author response:**

We thank the reviewer for this valuable comment. We acknowledge that fisheye photos are sensitive to sky conditions and exposure settings. During data collection, most images were taken at f/5.6 and 1/100 s. We always checked the pictures by experts in the field. When overexposure occurred, we increased the aperture value to minimize overexposure.

We could not standardize the time of image acquisition due to logistic constraints. Nevertheless, results remained consistent across conditions, indicating limited bias. We will discuss this limitation in the revised manuscript. L272:

"Fisheye imagery is particularly sensitive to sky conditions and exposure settings, which can alter gap fraction retrieval and thereby bias LAI estimates. Most images in this study were acquired under diffuse or moderately bright conditions using an aperture-priority mode (f/5.6, -1 EV compensation, 1/100 s exposure), with visual inspection by experienced operators to minimize overexposure. Nevertheless, occasional variations in illumination and exposure could have influenced canopy—sky classification, particularly under high-contrast sky conditions. While we attempted to maintain consistent settings, logistical constraints prevented complete standardization across sampling dates. Despite these potential limitations, the consistency of DHP–LT correlations across measurement dates suggests that exposure variability introduced only minor bias in our dataset. Future applications should consider capturing and processing RAW fisheye images to ensure linear radiometric scaling and reduce sensitivity to camera exposure. "

**Replies to comments from community #CC1 Hongliang Fang**

1) "The innovation of the manuscript is limited. This work closely follows another previous study by Liu et al. 2015b (doi:10.1139/cjfr-2014-0351). However, the previous paper simply performed empirical woody and clumping correction for deciduous broadleaf forest in order to match DHP LAI with litter trap observations. Such cite-specific adjustment is not generic.

The paper states that "it remains unclear if the DHP method enables to robustly track temporal LAI dynamics" (L4-5). Authors need to get familiar with current progress of using DHP for temporal LAI measurement. There are many related studies such as

(doi: 10.1016/j.agrformet.2014.08.005, doi: 10.1016/j.agrformet.2018.02.003) for seasonal crop LAI measurement with DHP. There are even many automatic DHP measurement studies:

https://doi.org/10.1016/j.agrformet.2022.108999

https://doi.org/10.1016/j.agrformet.2020.107944

https://doi.org/10.1111/2041-210X.14199"

**Author response:**

We thank the reviewer for this valuable and constructive comment. We agree that the empirical calibration approach applied here is **site-specific** and not intended as a generic correction. This is mentioned in Line 67-70 and 160-164, and we have now clarified this more explicitly in the revised text:

"The site-specific calibration aims to reduce local biases caused by stand heterogeneity and should not be considered a universal correction applicable to other forest types."

Additionally we also removed Line 174-175 to avoid the assumption that the model is generic and not site specific:

"This model can be applied for similar sites, particularly in areas with predominantly deciduous tree cover."

We fully acknowledge that several studies have demonstrated multi-temporal or automated DHP applications (e.g., Baret et al., 2010; Saitoh et al., 2012; Chianucci, 2020; and the recent automatic DHP works cited by the reviewer). However, our study differs in emphasizing multi-temporal **evaluation** rather than application; specifically, assessing how accurately DHP-derived LAI follows temporal litter-trap reference data under varying canopy and illumination conditions. While earlier works (e.g., Mussche et al., 2001 DOI: 10.14214/sf.575; Liu et al., 2015DOI: 10.1016/j.agrformet.2015.04.025) conducted repeated DHP-LT comparisons, these relied on earlier-generation or analog methods. Here, we extend this framework using **modern DHP processing with improved clumping correctionand image binarization** to evaluate DHP performance across the full leaf-fall season in a forest environment.

We have added the following clarification to the Introduction (end of paragraph starting at Line 55):

"While several studies have demonstrated multi-temporal or automated DHP acquisition for monitoring seasonal LAI dynamics (e.g., Demarez et al., 2014; Chianucci et al., 2018; Otsu et al., 2022; Zheng et al., 2020), only a few have compared DHP-derived LAI directly with repeated litter-trap observations over an entire leaf-fall period. For example, Mussche et al. (2001) and Liu et al. (2015) conducted such comparisons in deciduous and mixed forests, respectively, using earlier-generation or analog DHP methods. Here, we extend these efforts using modern DHP processing—featuring improved clumping correction, exposure control, and image binarization—to assess how well DHP-derived LAI captures temporal litter-trap dynamics throughout the leaf-fall season."

**2) "L19 Note that the "total intercepting area" is different from the flat area (L93). LAI is defined for the flat area, not intercepting area."**

**Author response:**

We thank the reviewer for the clarification. We will correct the definition of LAI to reflect that it is defined with respect to the *projected (flat) ground area*, not the total intercepting area. The revised sentence would be:

"As leaves constitute the active surfaces for these mass and energy exchanges, the leaf area index (LAI) emerges as a fundamental biophysical parameter, quantifying canopy structural complexity through its definition as half of the total leaf area per unit projected ground area [m²/m²]."

**3) "L100 For "the cumulative LAI", do you mean "the cumulative LT LAI"?**

**Author response:**

Thank you for pointing out this ambiguity. Yes, we were referring to the *cumulative LT LAI*. We will revise the text to specify this more clearly. The sentence would be:

"For each subsequent time step, the LT LAI was calculated by subtracting the cumulative LT LAI of all previous time steps from the total (Figure 2)."

4) "L130-138. The LXG method is essentially different from the LX method is the estimation of clumping index (Fang, 2021; doi: 10.1016/j.agrformet.2021.108374). The LXG CI is not a ratio of effective LAI to the true LAI."

**Author response:**

We thank the reviewer for this helpful clarification. We agree that only the LX method defines the clumping index as the ratio of effective to actual LAI, while the LXG method estimates clumping differently based on gap size distribution. We will make this distinction more clear in the revised manuscript and cite Fang (2021) in the intro to the clumping index.

**5) "L187. For Fig. 5 here, do you mean Fig. 4?"**

**Author response:**

We thank the reviewer for pointing this out. The reviewer is totally right, we will change this to Figure 4.

6) "Section 3.1. The comparison of DHP and LT LAI was not clearly presented. It's recommended to show a scatterplot to compare both DHP and LT LAI observations. Also show the effective LAI scatterplot and the clumping index derived from different view zenith angles."

**Author response:**

We thank the reviewer for this helpful suggestion. We included the comparison between DHP-and LT-derived LAI in Figure 6. Additionally, Figure 4 presents effective LAI across different view zenith angle (VZA) ranges in a summarized form to improve clarity and avoid excessive subplots. We will add further explanation of the clumping index derivation and its relationship with VZA in the revised manuscript.

7) "L191-194 can be moved to section 3.1. I guess the Fig. 3 in L193 should be read as Fig. 4."

**Author response:**

We thank the reviewer for the advice. However this sentence is more related to the temporal trend discussed in section 3.2. We will reformulate the sentence to make it more clear. Furthermore, we appreciate the reviewer's observation regarding the figure reference. This will be corrected from Fig. 3 to Fig. 4 in the revised manuscript.

**8) "Section 3.3. I would suggest to show the slopes and intercepts (Eq. (1)) for different phases."**

**Author response:**

We thank the reviewer for this helpful suggestion. We tested this and found that the slopes and intercepts differ only marginally, and including them would overcomplicate the figure. Showing the errors instead keeps the figure simple and easy to interpret. Therefore, in the revised manuscript, we report the errors for each phase in the figure for clarity.

**9) "Section 4.1. I would not use the term "spatial footprint of LTs" since footprint is mostly used for LiDAR observation in this community. LT data are supposed to represent the whole sample plot."**

**Author response:**

Thanks for mentioning that we exactly used this term due to the similarities to LIDAR methods. Because many experts in this field are familiar with this terminology. We asked several colleagues for their opinion and the general consensus was that footprint is an intuitive term. However, we are open for alternative suggestions.